# Effect of Cold Working on the Properties and Microstructure of Cu-3.5 wt% Ti Alloy

**DOI:** 10.3390/ma15228042

**Published:** 2022-11-14

**Authors:** Lue Huang, Lijun Peng, Xujun Mi, Gang Zhao, Guojie Huang, Haofeng Xie, Wenjing Zhang

**Affiliations:** 1School of Materials Science and Engineering, Northeastern University, Shenyang 110819, China; 2State Key Laboratory of Nonferrous Metals and Processes, GRIMAT Group Co., Ltd., Beijing 101407, China; 3GRIMAT Engineering Institute Co., Ltd., Beijing 101407, China; 4General Research Institute for Nonferrous Metals, Beijing 100088, China; 5China Nonferrous Metals Mining Group (Tianjin) Co., Ltd., Tianjin 300112, China

**Keywords:** Cu-Ti alloy, deformation, aging hardening, spinodal decomposition, strength, conductivity

## Abstract

Cu-Ti alloys were strengthened by β’-Cu_4_Ti metastable precipitation during aging. With the extension of the aging time, the β’-Cu_4_Ti metastable phase transformed into the equilibrium β-Cu_4_Ti phase. The Cu-3.5 wt% Ti(Cu-4.6 at% Ti) alloys with different processing were aged at different temperatures for various times after solution treatment at 880 °C for 1 h. The electrical conductivity of samples under different heat treatments had shown an upward trend as time increased during aging, but the hardness reached the peak value and then decreased. The hardness and electrical conductivity of the samples with 70% deformation after aging are higher tha n the samples without deformation. Deformation after aging would cause the metastable phase to dissolve into a matrix. The best combination value of conductivity and hardness is 13.88% IACS and 340.78 Hv, and the optimal heat treatment is 500 °C for 2 h + 70% deformation + 450 °C for 2 h.

## 1. Introduction

Compared with other alloys, copper alloys have high electrical conductivity even after alloying or heat treatment to improve mechanical properties. Therefore, copper alloys, such as wires and connectors, are excellent electrical materials. There are many ways to strengthen high-conductivity alloys, such as alloying, precipitation and grain refinement, but these strengthening methods lead to a decrease in conductivity [1,2,3]. For industrial applications of copper alloys, a balance between economy, mechanical properties and electrical conductivity should be considered, so the most effective method is precipitation hardening [4,5,6]. The well-known precipitation-strengthening copper alloys are Cu-Be [7,8], Cu-Ni-Si [9,10] and Cu-Ti [11,12,13,14] alloys. Cu-Ti alloy has high strength of 1 GPa and relatively high electrical conductivity compared with other copper alloys, so it is now used as connectors or conductive spring materials and has a wider range of applications as the increasing demand for high strength and high electrical conductivity [15,16,17,18]. Cu-Ti alloy undergoes spinodal decomposition during aging, forming the metastable phase β’-Cu_4_Ti (tetragonal structure with I4/m space group), which transforms into the stable phase β-Cu_4_Ti (orthorhombic structure with Pmmn space group) with prolonging aging time. During aging, the metastable phase β’-Cu_4_Ti precipitates within grains, and it transforms to stable β-Cu_4_Ti with a layered structure at the grain boundaries with the increase of aging time. The former is called continuous precipitation, and the latter is called discontinuous precipitation. Compared to other copper alloys, Cu-Ti alloy needs a longer aging time to obtain the proper combination of strength and electrical conductivity and has a lower electrical conductivity [19,20]. Therefore, it is necessary to introduce deformation processing, but the influence of deformation processing on the properties and microstructure of Cu-Ti alloy is still unclear. This paper focuses on the influence of deformation on the properties and microstructure of Cu-Ti binary alloy.

## 2. Experimental Procedure

Cu-3.5Ti (wt.%) ingots were obtained by smelting and casting pure copper and titanium in an intermediate frequency furnace with dimensions of 300 × 150 × 20 mm. The chemical compositions of the alloys were determined through inductively coupled plasma atomic emission spectrometry (ICP-AES), as listed in Table 1. Following the removal of the surface defects, the ingot was hot rolled at 850 °C from 20 to 5 mm. Thereafter, the hot-rolled samples of the Cu-3.5Ti alloy were solid solution-treated at 880 °C for 60 min to obtain a supersaturated solid solution. To confirm the optimal aging treatment, the alloy samples were aged at different temperatures at different times. The solid solution samples were isothermally aged from 400 to 525 °C for 1 h. The most suitable initial aging temperature and time are determined. After the initial aging treatment, the sample is deformed by 70% to 1.5 mm and then aged at different temperatures for a long time. The processing process diagram is shown in Figure 1. Subsequently, to remove surface defects, the ingot was milled to 1.2 mm. Moreover, the hardness of these samples, which were cut to the same size (10 × 10 × 1.2 mm) by wire cutting, was tested using a VH1150 Macro Vickers Hardness Tester with a 5 kg load for 15 s. Further, the conductivity tests of samples were conducted using an eddy current conductivity instrument Sigma 2008. Each sample was tested at least 5 times, and the values were averaged. The microstructure of samples with different aging times was characterized using a JEM-F200 field emission transmission electron microscopy (TEM) system. These samples were processed into 3 mm disks with 80 μm thickness followed by electrolytic double jet polishing from −40 to −50 °C using a solution of CH_3_OH:HNO_3_ = 70%:30%. Further, the X-ray diffraction (XRD) data were obtained using a SmartLab XRD instrument.

## 3. Results

Figure 2a shows the variation curves of the electrical conductivity and Vickers hardness of the Cu-3.5Ti alloy aged for 1 h from 400 to 525 °C. The electrical conductivity of Cu-3.5Ti alloys increased with the temperature rising during aging. The hardness of the alloy reached a peak value of 295.91 HV at 500 °C. Thereafter, the hardness of the alloy decreased with an increase in temperature over 500 °C, thereby indicating the occurrence of over-aging. As shown in Figure 2b, the optimum aging time at 500 °C is 2 h. Subsequent experiments will be conducted on samples with optimal aging.

The bright-field image and corresponding SAED pattern of Cu-3.5Ti at the peak aging state are shown in Figure 3. Long rod-like phases with a width of 26.4 nm are observed along the [001]Cu direction. From the SAED pattern, the diffraction patterns are confirmed as those of the β’-Cu_4_Ti phase in Figure 3b. The orientation relationship between the precipitation phase and the matrix can be obtained. The orientation relationships between the Cu matrix and β’-Cu_4_Ti phase were [001]Cu//[001]β’ and (100)_Cu_//(310)_β’_ at the peak aging time.

Figure 4 shows the bright-field image and corresponding SAED pattern of Cu-3.5Ti aged at 500 °C for 8 h. The discontinuous precipitates with lamellar structures are also found. From the SAED pattern, the diffraction patterns are confirmed as those of the β-Cu_4_Ti phase in Figure 4b. It indicates the formation of equilibrium discontinuous precipitates at the over-aging state. The orientation relationships between β-Cu_4_Ti and the Cu matrix were [110]Cu//[102]β and (111)_Cu_//(010)_β_.

Cold deformation was carried out on the samples in peak aging state. With the increase in deformation, the samples change from uniform deformation to non-uniform deformation, as shown in Figure 5. In the uniform deformation stage, the number of twin boundaries obviously increases, and in the non-uniform deformation stage, slip bands perpendicular to the rolling direction appear.

The samples with 70% cold deformation were aged at different temperatures and times, and their hardness and electrical conductivity were measured and shown in Figure 6. Taking the balance between conductivity and hardness into consideration, the best aging treatment of samples with 70% deformation is 450 °C for 2 h. Compared with the samples aged directly without deformation, the hardness and conductivity of samples with 70% deformation are both improved obviously, and the aging temperature is reduced correspondingly.

## 4. Discussion

The samples with different degrees of deformation after peak aging were tested by XRD. It was found that the peak corresponding to β’-Cu_4_Ti metastable precipitation gradually disappeared with the increased degrees of deformation, as shown in Figure 7. It indicates that for the deformed samples, the metastable β’-Cu_4_Ti phase had been dissolved, which is consistent with the significant decrease in the conductivity of the deformed sample from 12.62% IACS to 5.45% IACS.

The bright-field image of 70% deformed samples shows a large number of twin boundaries in the matrix, and the width of the β’-Cu_4_Ti phase decreases obviously, as shown in Figure 8. The twin diffraction spots can be seen from the SEAD pattern, but the spots of the metastable β’-Cu_4_Ti phase had disappeared, which also proves that the deformation after aging will make the β’-Cu_4_Ti phase dissolve into the matrix. Figure 9 shows the TEM image and element distribution maps of samples aging at 450 °C for 2 h after 70% deformation. The formation of the metastable β’-Cu_4_Ti phase can be clearly observed in Figure 9c, and the width of the precipitated phase is 16.3 nm, smaller and denser than the direct aging sample after solid solution treatment. Therefore, the property of the samples with 70% deformation is more excellent.

It can be concluded that the twin boundaries and slip bands generated by the deformation act as a heterogeneous nucleation site and accelerate the precipitation of β’-Cu_4_Ti, leading to an increase in hardness and conductivity at the same time. In addition, when combined with the change in hardness and conductivity according to the aging curve shown in Figure 6, the deformation before aging was able to effectively reduce the temperature to reach the highest hardness during aging. In this study, it has been found that aging with deformation processing is very effective for Cu-Ti alloys to improve both hardness and conductivity. It can be interpreted that Cu-Ti alloys are precipitated through spinodal decomposition. Spinodal decomposition occurs during aging, in which the solid solution is separated into a rich portion and a lean portion. This chemical composition forms a stress field in a microscopic range and inhibits the movement of dislocations or shear bands, which increases strength. Meanwhile, these defects act as heterogeneous nucleation sites. The precipitation of the β’-Cu_4_Ti phase is accelerated, and the β’-Cu_4_Ti phase is finer. It is interpreted as increasing the strength and conductivity within the alloy. As such, the aging with deformation processing in the Cu-Ti alloy reduces the precipitation temperature and increases the highest hardness. It can be an effective method to simultaneously improve strength and conductivity.

## 5. Conclusions

In this study, in order to examine the effect of the deformation of Cu-3.5wt% Ti alloy on aging, it was processed at 70% deformation after solid solution treatment and aged at 500 °C for 2 h. The change in hardness and electrical conductivity with aging was analyzed, and the reason was examined through microstructure analysis The following conclusions were drawn:
Cu-3.5Ti alloys were strengthened by the β’-Cu_4_Ti metastable phase during aging, with the extension of the aging time, the metastable phase transformed into the equilibrium β-Cu_4_Ti phase.Deformation after aging would cause the metastable phase to dissolve into the matrix, and the twins and slip bands generated during deformation would accelerate the precipitation of the metastable β’-Cu_4_Ti phase in aging treatment, reduce the precipitation of β’-Cu_4_Ti phase temperature of the precipitation phase, and make the β’-Cu_4_Ti phase finer.The optimal heat treatment is 500 °C for 2 h + 70% deformation + 450 °C for 2 h, the corresponding conductivity and hardness are 13.88% IACS and 340.78 Hv.

## Figures and Tables

**Figure 1 materials-15-08042-f001:**
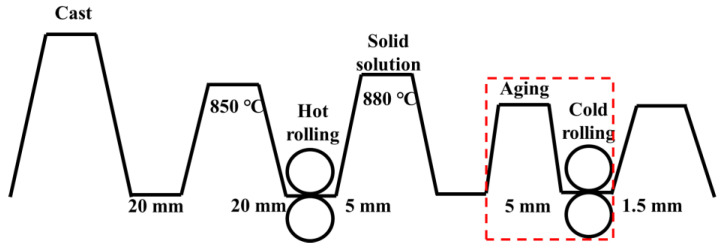
Schematic diagrams of thermo-mechanical treatment processes.

**Figure 2 materials-15-08042-f002:**
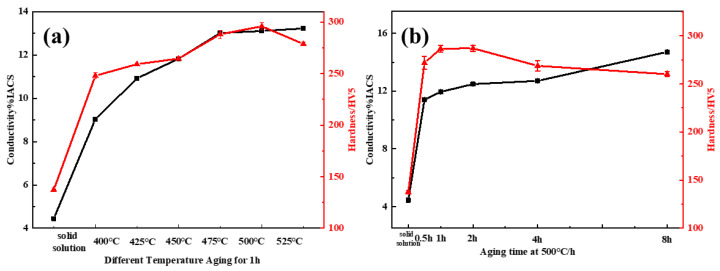
Hardness and electrical conductivity curves of the Cu-3.5Ti alloy aged at (**a**) different temperatures for 1 h and (**b**) 500 °C for different times.

**Figure 3 materials-15-08042-f003:**
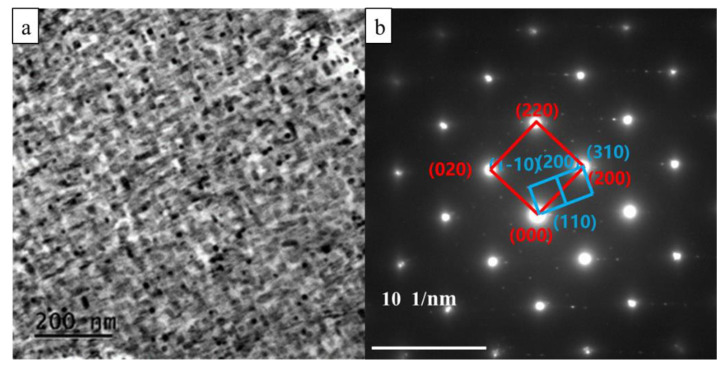
TEM and SAED images of the Cu-3.5Ti alloy at the peak aging state. (**a**) Bright-field image; (**b**) SAED pattern.

**Figure 4 materials-15-08042-f004:**
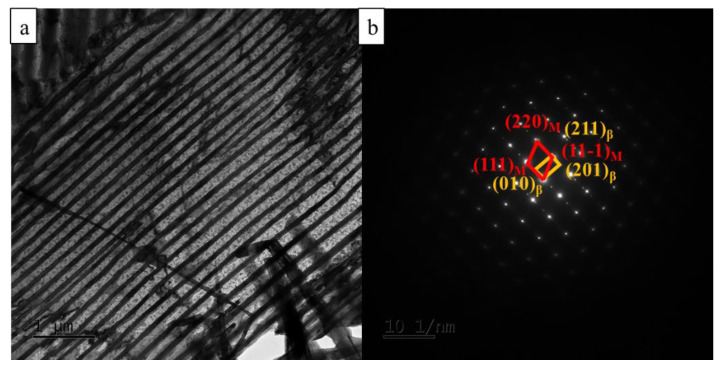
TEM images of the Cu-3.5Ti alloy aging at 500 °C for 8 h. (**a**) Bright-field image; (**b**) SAED pattern.

**Figure 5 materials-15-08042-f005:**
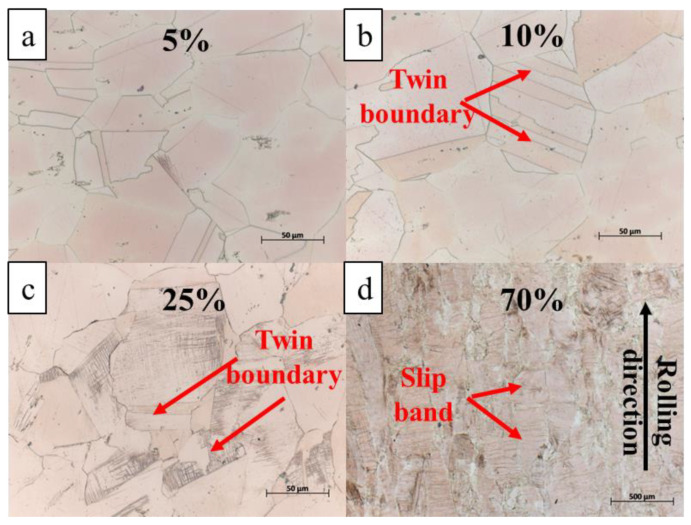
Optical photographs with different degrees of deformation (**a**) 5%; (**b**) 10%; (**c**) 25%; (**d**) 70%.

**Figure 6 materials-15-08042-f006:**
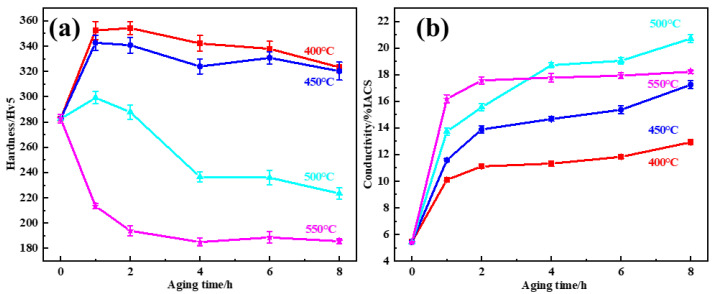
(**a**) Hardness and (**b**) electrical conductivity curves of the Cu-3.5Ti alloy aged at 400 °C to 550 °C with 70% deformation.

**Figure 7 materials-15-08042-f007:**
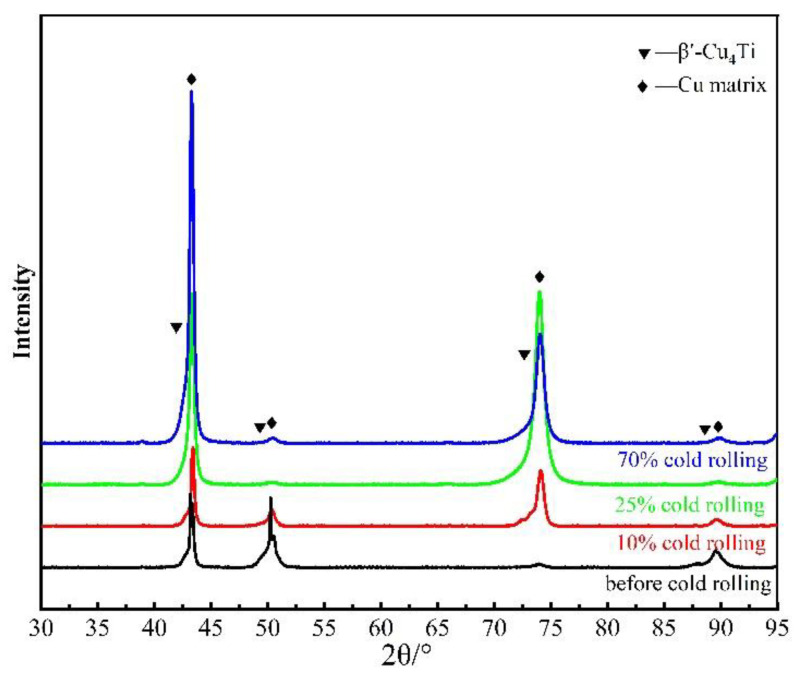
XRD patterns of samples with different degrees of deformation.

**Figure 8 materials-15-08042-f008:**
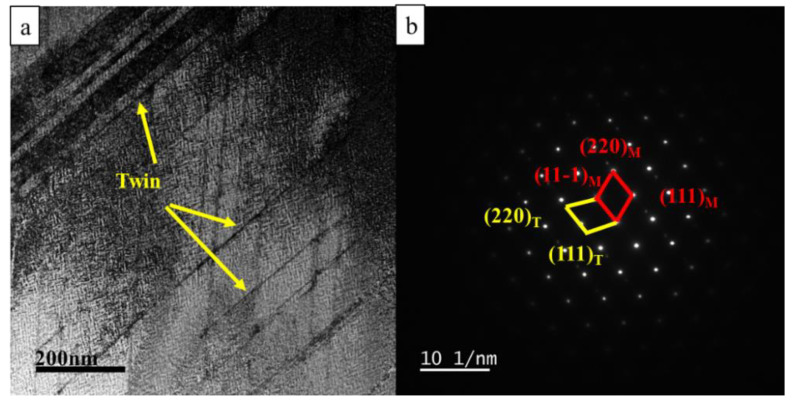
TEM and SAED images of the peak aging condition of Cu-3.5Ti alloy with 70% deformation (**a**) Bright-field image; (**b**) SAED pattern.

**Figure 9 materials-15-08042-f009:**
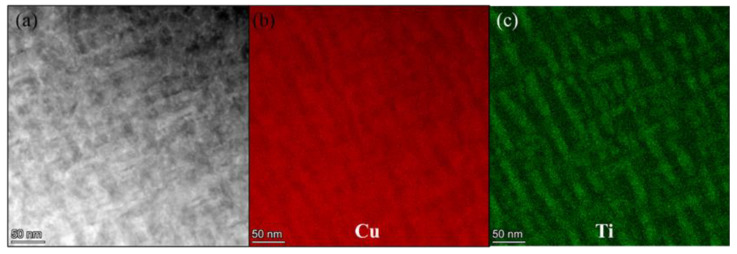
TEM image and corresponding element distribution (**a**) Bright-field image; (**b**) distribution of Cu; (**c**) distribution of Ti.

**Table 1 materials-15-08042-t001:** Chemical composition of the Cu-3.5Ti alloy (wt.%).

Nominal Composition	Ti	Cu
Cu-3.5Ti	3.58	Bal.

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
