# Peer review of "Effect of Cold Working on the Properties and Microstructure of Cu-3.5 wt% Ti Alloy"

_materials, 2022, doi:10.3390/ma15228042_

Round 1

Reviewer 1 Report

The paper by Liu Huang, Lijun Peng, Xujun Mi, Gang Zhao, Guojie Huang, Haofeng Xie and Wenjing Zhang "Effect of Cold Working on Deposition Behavior in Cu-3.5wt% Ti Alloy" is written with clear style and excellent presentation.

The problem is succinctly formulated and there is a specific understanding of the purpose of the article.

The only comment that can be made relates to the design of the text of the article (small letters after the dot, use of different font sizes). Check again the design of the bibliography.

Author Response

Response to Reviewer 1 Comments

Point 1: The only comment that can be made relates to the design of the text of the article (small letters after the dot, use of different font sizes). Check again the design of the bibliography.

Response 1: We appreciate for your valuable comment. We have checked our manuscript again, and capitalized the first letter after the dot, and unified the font size. The English language and style had also been checked. We feel sorry for our carelessness.

Reviewer 2 Report

This is a systematic investigation that the authors have performed on Cu-Ti alloy which is important from a technological perspective as well as to address a fundamental problem of circumventing the strength-electrical conductivity trade-off in metallic materials. The manuscript is well written but at times there is a slip; there is nothing like "more excellent" and the authors should avoid such faux pass. Some other detracting points are

1. Please discuss the interaction of different defects like twin boundaries and slip bands in scattering of electrons as it controls the electrical conductivity.

2. Remember the crystal structure of the solid solution phase is fcc, beta prime is bct and beta is orthorhombic, so use the correct brackets for showing orientation relationship. I see that the authors have used square and round brackets for cubic symmetry which is not really correct.

3. Provide quantitative estimate of microstructure in terms of dislocation density from x-ray diffraction; precipitate fraction and size distribution from TEM. This will increase the impact of the manuscript.

4. Provide a constitutive relation for strength from different mechanisms and relative contribution from different defects on electrical conductivity.

5. Explain the reasons for the improvement in electrical conductivity of rraged rolled samples in more detail.

Author Response

Response to Reviewer 2 Comments

Point 1: Please discuss the interaction of different defects like twin boundaries and slip bands in scattering of electrons as it controls the electrical conductivity.

Response 1: We appreciate for your valuable comment. Our experimental aim is to get the best deformation degrees before aging. The necking point (22.6%) was obtained through the tensile curve, so three different ratios (10% 25% 70%) were selected. It was originally speculated that uniform deformation would have the best improvement of the properties, but the test results show that the property of samples with large deformation is better than that of small deformation. Under the condition of peak aging, there is no difference between twin boundaries and slip bands for β´-Cu4Ti phase, but under the condition of over-aging, the slip band will accelerate the formation of β-Cu4Ti phase, while the twin will not, but this paper has not analyzed over-aging state.

Point 2: Remember the crystal structure of the solid solution phase is fcc, beta prime is bct and beta is orthorhombic, so use the correct brackets for showing orientation relationship. I see that the authors have used square and round brackets for cubic symmetry which is not really correct.

Response 2: In the manuscript, the orientation between precipitation and matrix was corrected, the corresponding diagram Fig 3 has also been corrected. and the use of brackets was unified. We feel sorry for our carelessness.

Point 3: Provide quantitative estimate of microstructure in terms of dislocation density from x-ray diffraction; precipitate fraction and size distribution from TEM. This will increase the impact of the manuscript.

Response 3: The dislocation density can't be calculated by XRD data, or the dislocation density calculated by XRD pattern can't be trusted, because the deformation will lead to precipitation phase dissolution and the distance between metastable phase and Cu diffraction peak is too close. The FWHM (full width at half maxima) measured is incorrect. We are also trying other methods to calculate dislocation density in alloys, such as TEM, but there's not much progress.

Point 4: Provide a constitutive relation for strength from different mechanisms and relative contribution from different defects on electrical conductivity.

Response 4: The same as the point 1, we are carrying out related work, but the corresponding results have not been sorted out and summarized.

Point 5: Explain the reasons for the improvement in electrical conductivity of rraged rolled samples in more detail.

Response 5: In the “discussion” part, some analysis about improvement of conductivity was added.

Reviewer 3 Report

October 19, 2022

Editor, materials

Title: “Effect of Cold Working on the Precipitation Behavior in a Cu- 3.5 wt% Ti Alloy” .

Manuscript Number: materials-1991370

Dear Editor,

        I am attaching my review comments of the manuscript on a paper entitled “Effect of Cold Working on the Precipitation Behavior in a Cu- 3.5 wt% Ti Alloy”.

.

In this paper, the authors have studied the influence of cold working (rolling) on the precipitation in a Cu-3.5 wt% Ti Alloy and on both the hardness and electrical conductivity. The microstructure observations were performed using TEM and optical microscopes. The results of the paper were well presented and discussed. It is exciting research; the reviewer suggests accepting this paper for publication in the materials after a minor revision to cover the following comments. 

1-      The paper's title must be correct to fit the actual forming process performed only in the last stage. The last stage of forming (cold rolling) can affect the size and distribution of precipitation not forming of them.

2-      The precipitation's type, composition, size, and distribution must be provided.

3-       In the experimental work, the last stage of forming, rolling, was performed under three different reaction ratios, as seen from the results XRD. However, in experimental work, there is no data about that. The experimental work needs more details; for example, until we reach the results part, we can still guess which conductivity authors measured, electrical or thermal.

4-      Where are the EBSD results used in obtaining the grain size? See line 71.

5-      XRD results in Fig. 7 need to be further explained, and phases must be added to the figure.

                                                                    Sincerely yours,

Author Response

Response to Reviewer 3 Comments

Point 1: The paper's title must be correct to fit the actual forming process performed only in the last stage. The last stage of forming (cold rolling) can affect the size and distribution of precipitation not forming of them.

Response 1: We appreciate for your valuable comment. According to your comment, we have changed our manuscript´s title to fit our content. The new title is “Effect of Cold Working on the Properties and Microstructure of Cu-3.5 wt.% Ti Alloy”.

Point 2: The precipitation's type, composition and size must be provided.

Response 2: The size of precipitation phase had been supplemented in this paper. In the “introduction” part, the type and composition of the two precipitations were described briefly. There is no difference in this experiment, so they are not introduced in the “results” part.

Point 3: In the experimental work, the last stage of forming, rolling, was performed under three different reaction ratios, as seen from the results XRD. However, in experimental work, there is no data about that. The experimental work needs more details; for example, until we reach the results part, we can still guess which conductivity authors measured, electrical or thermal.

Response 3: Our experimental aim is to get the best deformation degrees before aging under different reaction ratio. The necking point (22.6%) was obtained through the tensile curve, so three different reaction ratios (10% 25% 70%) were selected. It was originally speculated that uniform deformation would have the best improvement of the properties, but the test results show that the property of samples with large deformation is better than that of small deformation, so only 70% aging data after deformation were provided. As for conductivity, I had added electrical before conductivity in the manuscript.

Point 4: Where are the EBSD results used in obtaining the grain size? See line 71.

Response 4: We tested the sample by EBSD, but the relevant data was not used in this paper, so the sentence in line 71 was deleted. We feel sorry for our carelessness.

Point 5: XRD results in Fig. 7 need to be further explained, and phases must be added to the figure.

Response 5: We have modified the Fig. 7, and the corresponding diffraction peak of β´-Cu4Ti and copper matrix have been added to XRD results in the figure.
